# A New Database of the Quantitative Cathodoluminescence of the Main Quarry Marbles Used in Antiquity

**Philippe Blanc [1], M. Pilar Lapuente Mercadal [2,3],*  and Anna Gutiérrez Garcia-Moreno [3]**

1   UPMC—Faculté T.E.B., Sorbonne Université, 4 place Jussieu, CEDEX 05, 75252 Paris, France; philippe.blanc64@sfr.fr
2   Department of Earth Sciences, University of Zaragoza, C/Pedro Cerbuna, 12, 50009 Zaragoza, Spain
3   Institut Català d'Arqueologia Clàssica (ICAC), 43003 Tarragona, Spain; agutierrez@icac.cat
*   Correspondence: plapuent@unizar.es; Tel.: +34-976-762125

**Abstract:** Quantitative cathodoluminescence (CL) has rarely been applied for the archaeometric studies concerning marble provenance, despite its potential. This paper develops the method and provides a new database of the parameters obtained from the main marble quarries used in antiquity. With a total number of 473 marble samples from ten districts of the central and eastern Mediterranean, it is the first database on quantitative CL, with the additional advantage of being the same samples that have already characterized by other conventional techniques and that are available in the literature. Focused on the measurements of the intensity peaks at the UV and visible spectra, registered by a spectrometer coupled to a scanning electron microscope (CL-SEM), the representative values are plotted on different useful diagrams to be applied in the identification of marble provenance studies, as a complementary tool of other analyses.

**Keywords:** archaeometry; marble; provenance; cathodoluminescence; CL-SEM

---

## 1. Introduction

Carbonates and other minerals emit radiation in the form of photons when they are bombarded by a source of energy; the process of luminescence is called cathodoluminescence (CL) since the excitation source is a beam of high-energy electrons emitted by a cathode. Fundamental causes of CL in minerals are widely explained in literature [1–4].

In general terms, during the process of electronic excitation, each unpaired electron in the crystal structure is promoted to a higher electronic level but, after a few microseconds, it returns back, emitting a photon of a defined wavelength. The emission of these characteristic photons is in the range of visible luminescence, but also in the UV radiation. A priori, the greater the number of electron traps (referred to as luminescence centers) present as impurities (extrinsic centers) or as lattice imperfections (intrinsic centers) in the crystal structure, the greater will be the CL intensity [4]. In most cases, the luminescence centers are ions inserted into the structure as impurities in trace element concentration. In fact, it is well known that in carbonate minerals $Ca^{2+}$ and $Mg^{2+}$ can easily be replaced by $Mn^{2+}$ promoting CL [5–8].

In calcite, this effective activator center emits at 620 nm, a visible radiation equivalent to a range of colors from yellow to orange. Nevertheless, other defects in the structure give intrinsic emissions in the field of UV, which seem to be related to the very dark blue luminescence cited in the literature [2]. On the other hand, dolomite moves luminescence to 650 nm, hence exhibiting a visible red luminescence. Although the presence of other ions can also influence the CL emission, activating (REE) or inhibiting ($Fe^{2+}$, $Ni^{2+}$ and $Co^{2+}$) among others, there is a good correlation between

the concentration of Mn in these minerals and the effect of luminescence expressed in terms of intensity [2,4]. Consequently, in marble, the CL characteristics are a function of the relative abundance of activators/quenchers present in carbonate from which $Mn^{2+}$ is the most relevant impurity, but also the carbonate crystallization and metamorphic recrystallization might cause structural defects in the crystalline lattice of natural carbonate.

Two main instruments and techniques benefit from the CL phenomenon, optical microscopy, and scanning electron microscopy (SEM). The first is based on the observation of standard thin-sections, and the visible radiation is recorded as microphotographs, named CL-imaging or CL-patterns. The equipment is composed of a standard petrographic microscope with a coupled CL stage, including a small vacuum chamber and a cold-cathode electron gun. In the second one, a spectrometer coupled to an SEM registers the spectra and measures the intensities of the peaks.

Since the pioneering work by Renfrew and Peacey [9], a significant number of research papers have focused on the practical use of CL in carbonate as a complementary tool for the provenance study of ancient marbles [10–20]. Most of them use representative CL-images of quarry marbles to identify archaeological pieces as a complementary technique to the usual C and O stable isotopes. By this method, three parameters can be qualitatively characterized: color, intensity, and distribution of CL [21].

On the contrary, very few papers address the issue of the application of quantitative CL to archaeometric studies. Until now, quantitative-CL has been approached in a tentative way, with a limited number of quarry samples characterizing either the classical marbles [22–25] or the main Hispanic ones [26,27]. As in both cases, quantitative CL was very promising, with a successful application to marble provenance of archaeological samples [28,29], Now a new quantitative CL database is reported here as the main aim of this paper. Focused on the quantification of the spectra intensities obtained by CL-SEM in a large number of samples from the most important marble quarries of the central and eastern Mediterranean, it constitutes the first database of these features. It is worth noting that all samples examined in this study belong to other published databases that compile the results of other analytical techniques [20,30,31]. This fact increases the value of the work since it provides additional parameters to other characteristics, such as petrographic features, the maximum grain size (MGS), or compositional factors such as their C and O stable isotopes, and Electron Paramagnetic Resonance (EPR), among others. The representative CL intensities in each marble site were obtained and projected on graphs that will be helpful for future applications in multi-method identification of marble artifacts. As a secondary but complementary objective, the paper focuses on the comparison between the CL-microphotographs obtained in two different laboratories using the same thin-sections and two different equipment systems, highlighting the convenience of completing the usual qualitative CL observations with the quantitative measurements, like those reported here.

## 2. Materials and Methods

A total of 473 samples from the main quarry districts in the central and eastern Mediterranean areas were measured, and their representative CL values are shown in Table 1. Their respective locations are available elsewhere [20,30,31]. The whole CL database is available as Tables S1–S10, accessible on-line. Samples were provided by D. Attanasio, M. Brilli, and M. Bojanowski, to whom we acknowledge their generosity.

An aluminum plate of the size of the sampling window in the CL-SEM was designed and enabled with a series of 30 small holes destined to be filled with the powder sample (3 mg each). The pieces and the necessary steps to prepare the highly compressed tables are illustrated in Figure 1.

A CL spectrometer accessory for an SEM was used to register the CL emission spectra and measure the characteristic peak intensities. A Zeiss Supra 55 VP SEM device was used, developed at the laboratory of the Université Pierre et Marie Curie (Sorbonne, Paris, France), which consists of a parabolic mirror as collector coupled with fiber to a grating spectrometer Triax of Jobin and Yvon with a cooled Liquid Nitrogen Temperature (LNT) and CCD camera. The system is able to operate

from 250 to 1200 nm. Because of the Vacuum Partial function, it is not necessary to coat the samples. Measurements were performed between 300 to 750 nm by means of 3 spectra taken in 1 s each, on three places of the same compressed powder, and the analytical conditions were 20 kV, 40 nA, under 20 Pa vacuum. CL emissions (cps) were noted at 620 nm or 650 nm and at two main UV peaks after the experimental test done in one Carrara sample (G13) explained in Section 3.1.

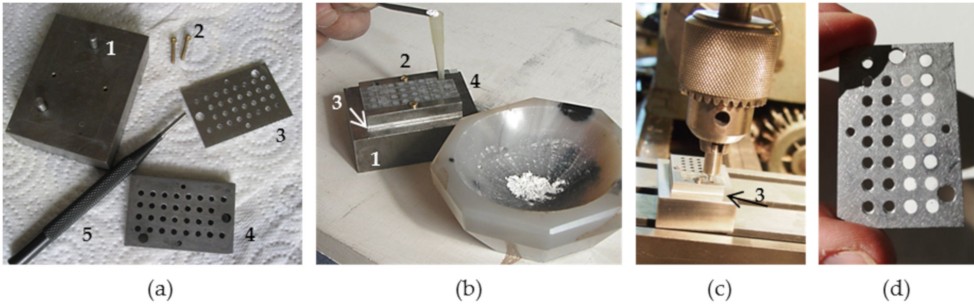

**Figure 1.** The sequence of the sample preparation: (**a**) pieces to be used; a small platform (1), two screws (2) to adjust the perforated aluminum plate (3) in which the samples will be inserted and placed under the perforated plate (4) and a manual piston (5) to help introduce the powder sample in each hole. (**b**) After grinding, 3 mg of powder sample is introduced in each hole; (**c**) each sample must be highly compressed; (**d**) example of an aluminum holder plate with half of the prepared tablets.

A set of classical marble thin-sections were observed under a cold cathode equipped optical microscope, and CL-images were recorded. At the Parisian laboratory, a Cathodyn device delivered by OPEA coupled to a standard Nikon microscope was used under the following analytical conditions (vacuum 0.08 mbar, beam current 250–300 µA, electron energy 12–15 kV and automatic or manual exposure time (from 1/6 s to 5 s, ISO 500).

Another cold equipment, a Technosyn CL8200Mk5-1, provided by CITL of Cambridge (UK) coupled to a Nikon Eclipse 50iPOL microscope, at the laboratory of ICAC in Tarragona was also used to take CL-microphotographs with an automatic digital Nikon Coolpix5400 camera. The electron energy applied was 15–20 kV, the beam current operated at 250–300 µA, while the vacuum was 0.17 mbar (17 Pa). In this case, the CL-images were automatically recorded (29 mm focal length, f/4.6 aperture, 1 s exposure time, ISO 200).

## 3. Results and Discussion

To obtain representative values for each quarry, in terms of quantitative CL intensity of the registered peaks of the spectrum, a first experimental assay was carried out on one marble sample, which is explained in Section 3.1. Once the operational process for obtaining the spectra was completed, and their representative peaks with their intensity measured, different graphs of quantitative CL were projected in Section 3.2, where according to their MGS different representation plots of the CL intensity quantification for all classical marbles were proposed. Additionally, to show the potential utility of the quantitative CL data for future archaeometric studies, certain CL values were combined with C and O isotopic data in different binary diagrams. Finally, Section 3.3 explores, in a complementary way, the comparison of the CL-patterns taken in the two different aforementioned laboratories, knowing that certain differences could be obtained [20,21].

### 3.1. Experimental Assay on a Marble Sample

A first test was conducted using the CL-SEM detector on one sample (G13 from Carrara) chosen because of its intensive emission in both UV and at 620 nm (Figure 2a). In detail, the presence of two large peaks around 350 and 400 nm can be observed (Figure 2b), as well as the strong emission around 620 nm. When this sample was bombarded at 20 kV, 40 nA in a partial vacuum of 20 Pa, under a rapid scan in an approximate area of 2 × 3 µm, no change, from one spectrum to the next, was observed over

time. On the other hand, when the beam was concentrated in a fixed spot, the peak around 350 nm decreased, while that close to 400 nm increased in intensity over time.

Sequential measurements at different times (0, 1, 5 and 10 min) under continuous bombardment are shown in Figure 2c. It should be noted that the extrinsic intensity of emission at 620 nm is almost not modified, while the intensities of the intrinsic peaks at 355 and 396 nm in the UV are sensitive to the deformations experimented in the calcite structure. To reduce the significant noise of the spectrum, a sliding average was applied (based on 10 points) from the beginning of the spectrum to its end. However, this had the disadvantage of shifting the peaks in the order of 2 nm, but starting again from the end to the beginning brought the spectrum back to its initial position, while considerably reducing the noise.

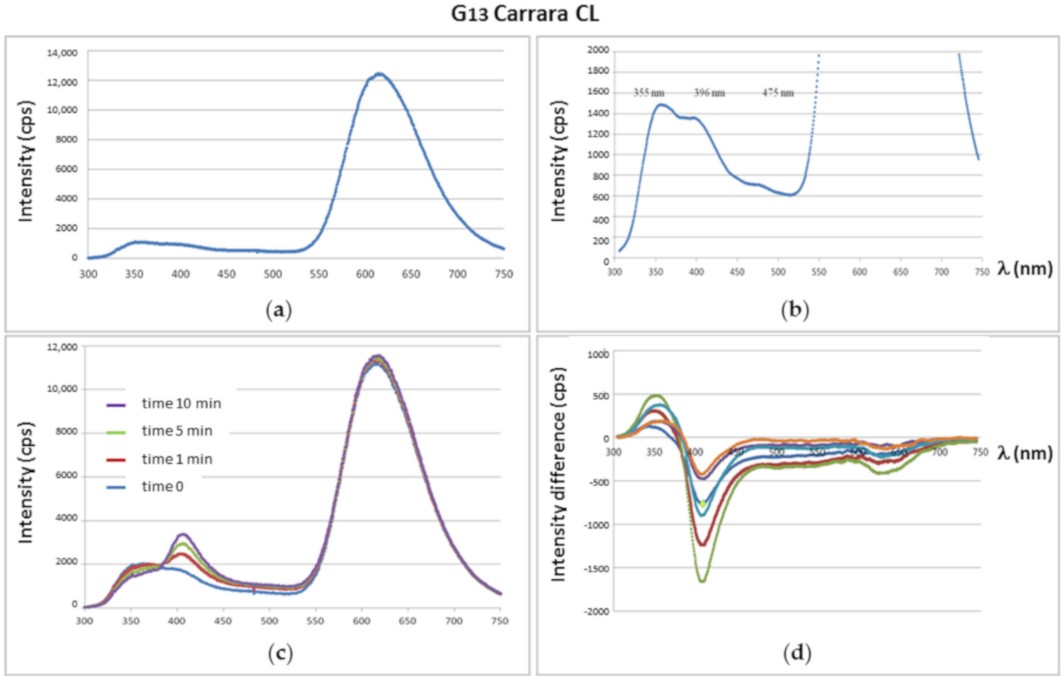

**Figure 2.** Experimental assay on a marble sample from Carrara (G13): (**a**) General CL emission spectrum; (**b**) detail of the UV peaks at 355, 396, and 475 nm; (**c**) temporal evolution of the spectrum under fixed spot and continuous bombardment; (**d**) associated representation of the differences between the successive spectra over time.

In Figure 2d, the positions of the peaks at 352 nm (±8 nm) and 408 nm (±2 nm) are highlighted, while that at 475 nm, shown in Figure 2b, goes now unnoticed. Note that at the 620 nm position, there is a less significant difference than those of 352 nm and 408 nm positions, which have precise and measurable variations. These UV emissions are the intrinsic emissions of calcite, which are modified by bombardment and represent a significant amount of energy.

Consequently, besides the energy emitted by the free electron of $Mn^{2+}$ in the visible spectrum, this part of UV energy should also be used to characterize any marble sample and represent them on a diagram. Thus, to represent the luminescence of each marble sample, the intensity of both UV peaks will be measured and expressed by their sum, versus the intensity of the peak due to the $Mn^{2+}$ impurities. At the same time, it must be mentioned that by measuring these intensities strictly at 352 and 408 nm, the small emissions due to the nitrogen present in the chamber with a partial vacuum (20 Pa) do not affect the measures.

### 3.2. Quantitative CL Diagrams of the Main Quarry Marbles

After a careful spectral processing carried out in all samples, to avoid the spectral shift and to limit the noise of measurements in each sample, two emission values of intensities (X, Y) allow

building quantitative CL emission diagrams. The X-axis refers to the measurements at the 620 nm peak, except for the dolomitic samples of Thassos (650 nm), while in the Y-axis, the sum of the intensities of intrinsic CL is displayed (Figures 3–7). Table 1 reports the representative values for each quarry site where, in additions to the maximum and minimum values, other statistical data are shown: Q1 (25th percentile), Q2 and Q3 (75th percentile) quartiles and the outliers defined as values 1.5 times out of the interquartile (Q3–Q1) range. This interquartile provides information about the statistics spread of the data, while Q2 (50th percentile) is the median of data. To show the differences of dispersion in each X and Y axis, not only all samples for each quarry were plotted in their respective graph, but also their lineal representation was automatically calculated, whose slope (*a* value) is also significant since shows which of the two spectral intensities (intrinsic or extrinsic) presents a greater dispersion. One example is illustrated in Figure 4, where Carrara and Afyon quarries are compared in the same diagram. Other *a* values are listed in Table 1.

However, aware of the need to present graphs to aid the archaeometric identification studies, the data has been separated into two groups. One includes fine-grained marbles (Afyon, Pentelicon, Carrara, Göktepe, and Paros−1) with MGS < 2 mm, and the second those of medium to coarse-grained marbles (Thassos dolomitic and calcitic, Naxos, Paros (2–3), Proconnesos and Aphrodisias) with MGS > 2 mm. The analytical values of both groups have been plotted in two different diagrams (Figures 8 and 9). Figure 8a,b show 90% probability ellipses X-Y diagrams and Figure 9a,b include all data (even outliers) using the usual boxplots and whiskers for each X and Y-axis.

**Table 1.** Quantitative CL of the main marbles used in antiquity, expressed by the intensity statistics values of the spectra. X (peak at 620–650 nm, or extrinsic CL) and Y (sum of peaks at UV, or intrinsic CL). Minimum (Min), maximum (Max), Q1 (25th percentile), Q2 (median), Q3 (75th percentile). Outliers (values 1.5 times out of the interquartile range). The slope (*a*) values are obtained from the linear representation of all samples in each quarry. They are listed in decreasing order of their median extrinsic intensity.

| Quarry (n. Samples) | X (Extrinsic CL) | | | Y (Intrinsic CL) | | | Linear (y = ax) a Value |
|---|---|---|---|---|---|---|---|
| | Min/Max (Outliers) | Median Q2 | Statistics Dispersion Q1/Q3 | Min/Max (Outliers) | Median Q2 | Statistics Dispersion Q1/Q3 | |
| Afyon (66) | 864/64,511 (>54,919) | 16,374 | 10038/27,990 | 682/2321 (>2269) | 1360 | 1044/1534 | 0.0418 |
| Thasos dol. (34) | 4201/55,733 (<33,936) | 9858 | 6877/17,701 | 684/1651 | 1104 | 930/1258 | 0.028 |
| Pentelicon (58) | 2668/35,231 (>24,479) | 9864 | 7000/13,992 | 269/2007 (<278) | 1118 | 941/1383 | 0.0686 |
| Naxos (34) | 2209/24,665 (<638) (>12,679) | 6562 | 5153/8164 | 265/921 | 517 | 408/630 | 0.0539 |
| Carrara (110) | 939/12,475 (>10,687) | 4071 | 2523/5789 | 618/7407 (>6667) | 2924 | 1971/3849 | 0.5484 |
| Thasos calc. (38) | 788/32,298 (>13,939) | 4065 | 3086/7427 | 316/1259 | 771 | 584/906 | 0.046 |
| Paros 2–3 (20) | 1183/6155 | 3348 | 1603/4799 | 786/903 | 843 | 809/878 | 0.2067 |
| Göktepe (19) | 353/5504 (>3317) | 1919 | 639/4399 | 848/2928 | 1919 | 1406/2655 | 0.582 |
| Paros−1 (9) | 1195/3513 | 1918 | 1571/2270 | 832/937 (<749) | 871 | 839/899 | 0.3965 |
| Aphrodisias (34) | 320/5510 (>2432) | 744 | 616/1343 | 43/2748 (<63; >1814) | 878 | 720/1158 | 0.3791 |
| Proconnesos (51) | 277/1867 (>1324) | 618 | 484/820 | 259/1058 (>595) | 433 | 410/484 | 0.5207 |

### 3.2.1. Quantitative CL of Carrara Marbles

Located in the NW of the Apuan Alps, North Italy, Carrara is one of the most famous marble extraction sites. Three main districts are represented in the database: Torano (Polvaccio,

Sponda, and Ravaccione), Miseglia (Canalgrande and Fantiscritti), and Colonnata (Fossacava, Calagio, and Gioia). Measurements carried out on 110 samples are very variable in both intensities related to the $Mn^{2+}$ peak (620 nm) and the one corresponding to the sum of (UV peaks), or intrinsic CL. Drawn in a binary diagram (Figure 3) a strong dispersion becomes apparent, as much in the X-axis (from 939 to 12,475 units, being outliers those values higher than 10,687), as in Y (from 618 to 7407 units, being outliers those higher than 6667) The interquartile goes from 2523 to 5789 in X, and from 1971 to 3849 in Y.

Indeed, with respect to the total set of data, among all the analyzed marbles, Carrara samples are also those with a wide dispersion in both axes, but in particular, in the Y-axis, whose median value (2924) is the higher median from all considered marbles (Table 1). However, the median value in the X-axis has an intermediate value (4071) regarding the median of all marbles (ranging from 618, in Proconnesos, to 16,374, in Afyon).

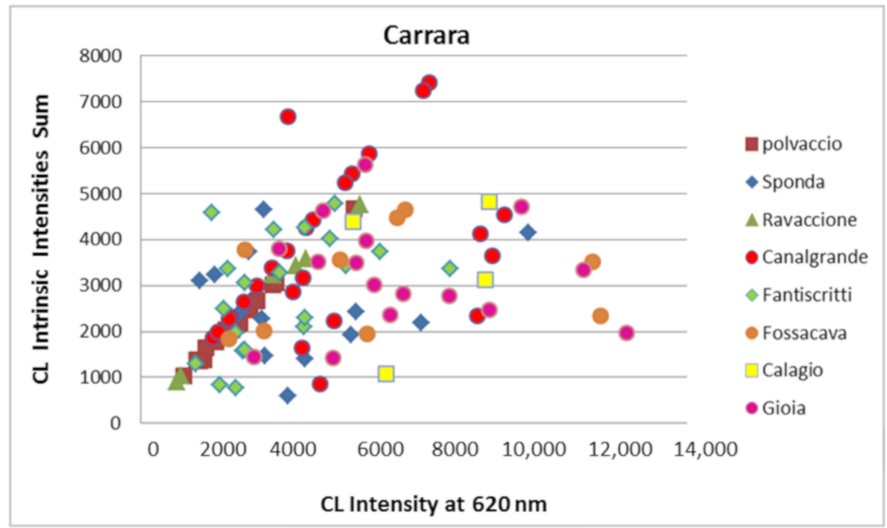

**Figure 3.** Quantitative CL diagram for Carrara marble from different quarries located in three valleys (Torano, Miseglia, and Colonnata, according to [30]).

### 3.2.2. Quantitative CL of Afyon Marbles and Its Comparison with the Quantitative CL of Carrara

Sixty-six samples of white marble from Afyon (Docimium), the modern city of Iscehisar in Asia Minor, were analyzed and plotted in Figure 4. Measurements obtained for the UV spectra (Y-axis) are relatively constant, ranging from 682 to 2321 units, with outliers above 2269 (Table 1). However, the very wide dispersion on the extrinsic CL intensity (X-axis), spanning more than 60,000 units, is evident, with outliers higher than 54,919 units. This feature is represented by the gentle slope of its linear representation (y = 0.0418x) in Figure 4. On the contrary, Carrara samples show an opposite luminescent behavior, with a high slope in its linear representation (y = 0.5484x), shown in the same figure. In fact, both examples are among the classical marbles with the greatest variation in both axes, respectively, as can be seen in the range of the interquartile values for both X and Y axes (Table 1 and Figure 5). This clear difference in CL behavior in both types of marble highlights the importance of plotting X vs. Y.

### 3.2.3. General Quantitative CL Diagrams for the Main Classical Marbles

In addition to the Turkish samples from Afyon, and following the order in Table 1, samples of the Greek marbles from Thasos (dolomitic and calcitic) and Naxos islands, as well as those from the Mount Pentelicon, near Athens, emit a wider range of CL radiation on the visible spectrum than those from Carrara (Figure 5), but all have lower intensity on the UV peaks than those of Carrara. Thasos calcitic marble has a median value in the X-axis quite close to that of Carrara (Table 1).

The joint projection in Figure 5 of all data from each quarry site enables not only the comparison between and among the diverse marble sites but also illustrates how wide the dispersion is of both CL modes (intrinsic or extrinsic) for each marble, an aspect related to the slope of their corresponding linear representation (Table 1). In particular, two different CL behaviors might be distinguished from this graph, which seems to follow similar guidelines to those described for Carrara and Afyon in Figure 4. So, the lineal representations for Afyon, Pentelicon, Naxos, Thasos calcitic, and dolomitic marbles have gentle slope values in response to the wide dispersion on their extrinsic CL intensities and in relation to their more restrictive intrinsic intensities. On the contrary, the steep slope of the linear trends of Carrara, Göktepe, Proconnesos, and Aphrodisias marbles indicates that they have a relatively greater dispersion in UV intensities than in those corresponding to the visible spectrum.

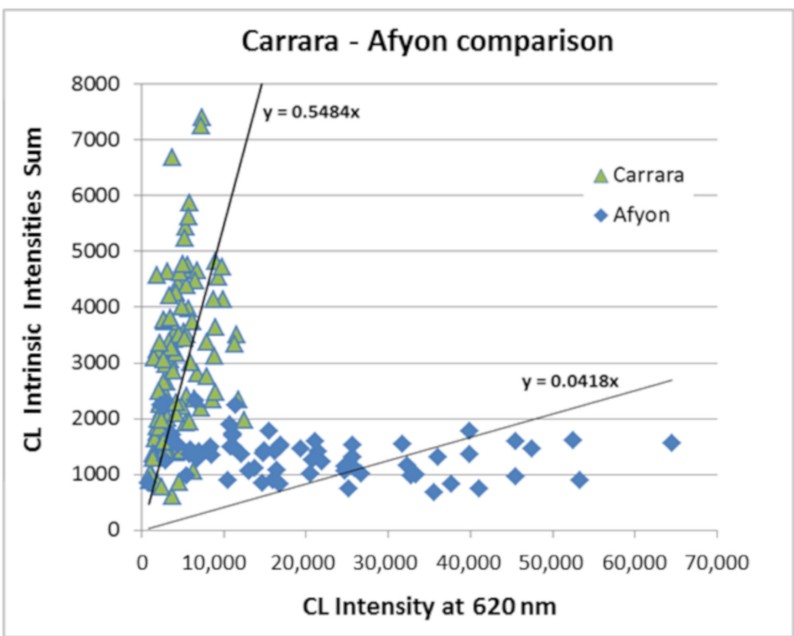

**Figure 4.** Comparison between Carrara and Afyon quantitative CL data and their respective linear representations.

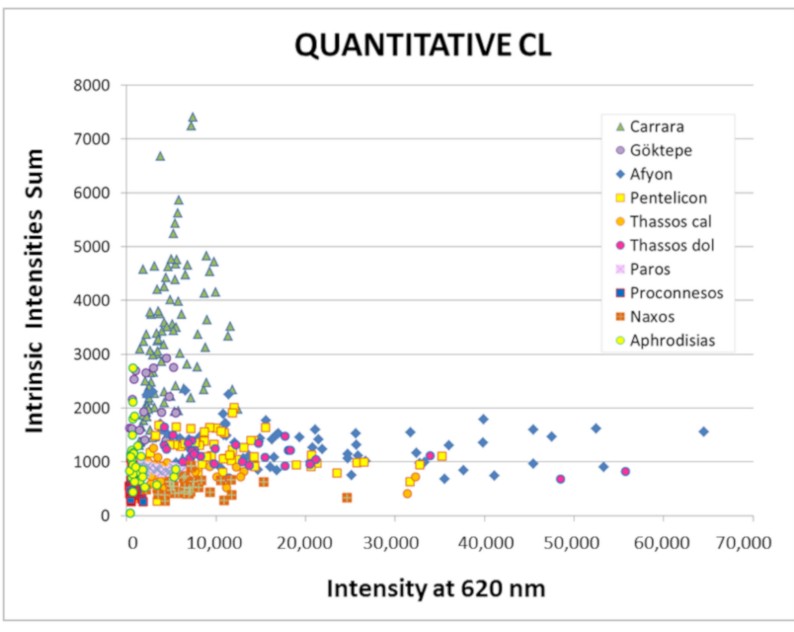

**Figure 5.** General quantitative CL diagram for the main classical marbles.

On the other hand, it is obvious that the general CL representation of all samples under consideration (Figure 5) shows a high degree of overlap, especially in marbles with low-intensity values for both variables. In detail, Figure 6 illustrates only those classical marbles with low intrinsic and extrinsic CL intensities. They are the Turkish marbles of Proconnesos (Marmara Island), Aphrodisias, and Göktepe, along with all marbles analyzed in the Cycladic island of Paros. This is not the case of Carrara marble, as seen above. However it is known that some Carrara marbles, especially certain Fantiscritti samples, also exhibit low visible radiation. This aspect would be worth treating in-depth since it is admitted that Fantiscritti marbles, analytically very different from other Carrara marbles, were used in emblematic ancient artistic works [30]. Although this topic is outside the scope of this paper, in the following Section 3.2.4 we will try to compare those samples of Fantiscritti to Göktepe samples, confronting their low extrinsic CL intensity with other parameters measured in the same samples taken from the published databases [20,30], as one example of the quantitative CL potential.

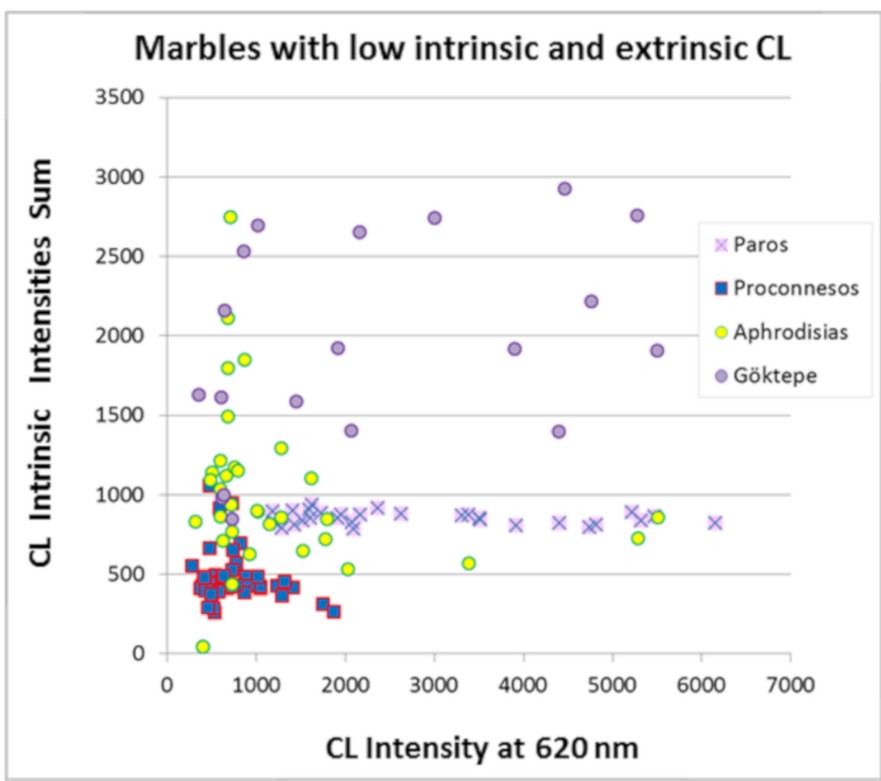

**Figure 6.** Quantitative CL diagram for classical marbles with low intensity in both axes.

Some relevant features must be highlighted in Figure 6. First, among all the analyzed samples, Proconnesian marble manifests very homogeneous values and the lowest intensities in both intrinsic and extrinsic CL. This behavior must be related to its homogeneous dark blue CL-pattern, which will be discussed in Section 3.3. As for the Aphrodisian marbles, certain samples share very similar values to those of Proconnesos (at both CL intensities), aspect of additional interest since this fact is in agreement with the CL-images recognized in Aphrodisias marble [16].

Paros and Göktepe marbles require a specific diagram for each of them to help to interpret their CL characteristics. On the one hand, it is well known that in Paros Island, several varieties of marbles occur, all of them of great archaeometric importance, which were extracted from quarries located in different valleys. For this reason, lychnites marbles, commonly known as Paros−1 in the usually isotopic diagrams, have been distinguished from those cropping out in Chorodaki or Marathi (*non-lychnites*) areas. This distinction is also necessary here since their respective petrography, and in particular MGS are different. Thus, in Figure 7a both Parian marbles are represented separately. Lychnites samples, in general, exhibit lower CL intensities than those from Chorodaki areas, in agreement with

the qualitative estimation of intensities in their respective CL-pattern (see Section 3.3). However, it is striking that the set of Parian samples shows similar very homogeneous intrinsic CL values, practically describing a straight line parallel to the X-axis. This CL behavior is unique and allows for relatively reliable identification as can be seen in Figure 9b.

Due to its importance in sculpture, the recently discovered Göktepe marble has received much attention from different scholars. Thus, it has been the subject of various publications aiming to discriminate it from other fine-grained marbles [18,20,29,31–33]. The scarce analyzed samples in this contribution have been plotted separately into two subgroups in the CL diagram (Figure 7b), according to who supplied each set of samples. In this particular case, three black and three grey quarry marbles were also included in the analyses, which are drawn with their corresponding colour in the CL diagram. Although some differences are detected between both subgroups, they are minimal, considering the total set of data. What could be significant is the dispersion on the extrinsic CL values, ranging from 605 to 5276 in the white samples, which inform that differences on the visible CL-patterns are possible in agreement with the slight discrepancies qualitatively detected in their respective CL-microphotographs reported in two different papers [18,20].

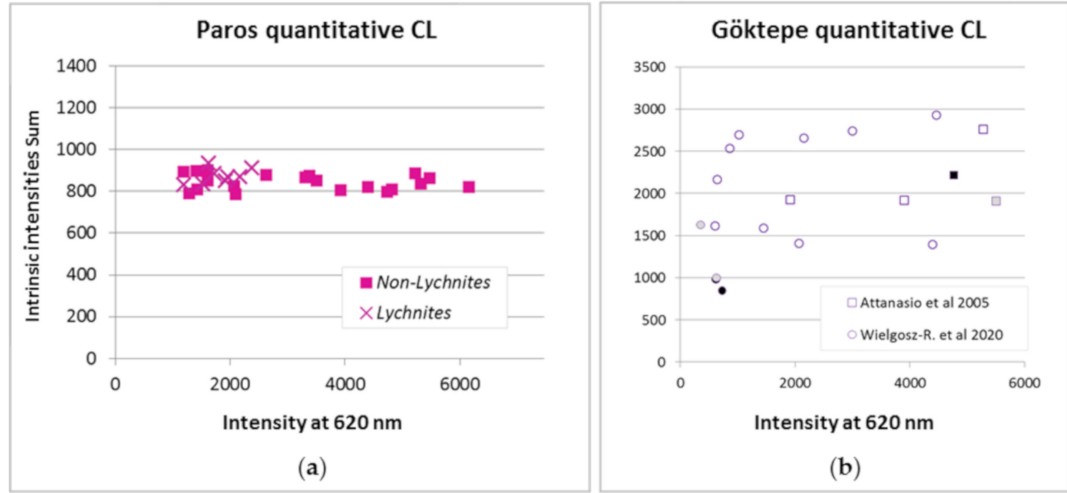

**Figure 7.** (**a**) Quantitative CL diagram for Parian marbles, Paros−1 (lychnites) and non-lychnites (Paros 2–3) according to their location [30]; (**b**) Quantitative CL diagram for Göktepe marbles with an indication of providers. Three samples of black Göktepe (in black symbols) and three grey marbles (in grey symbols) were measured.

As previously mentioned, using the MGS as the first discriminating parameter, we have considered appropriate to show the data set of all quarry marbles separately in two groups to facilitate its future application in multi-method archaeometric studies of provenance. The analytical values of both groups have been plotted in two different diagrams (Figures 8 and 9). Figure 8 shows 90% probability ellipses X-Y in two diagrams (Figure 8a,b). The use of this presentation type, quite usual for isotopic data, tries to minimize the degree of overlapping, which is evidently less than that shown in Figure 5. Each figure shows the level of discrimination attainable for the fine-grained (Figure 8a) and the medium to coarse-grained marble varieties (Figure 8b).

However, aware of the importance of not deleting data, even if they are outliers, we have also opted for the representation of all data in the typical boxplots and whiskers graphics for each intrinsic and extrinsic CL (Figure 9a,b).

The boxes of Figure 9 emphasize how the statistics ranges of quantitative CL appear to be promising as complementary markers for provenance studies. It is obvious that taken individually each variable may seem useless because all show overlapping to some extent. In particular, the restrictive values of Proconnesos, and those of Parian marbles as well as some Carrara and Göktepe marbles seem to be favorable for their discrimination although overlap is also observable in both, intrinsic and

extrinsic CL. In this regard, it should be remembered that there is no single discriminating technique for marble identification, so it is always necessary to apply a multi-method approach (See the example shown in Section 3.2.4).

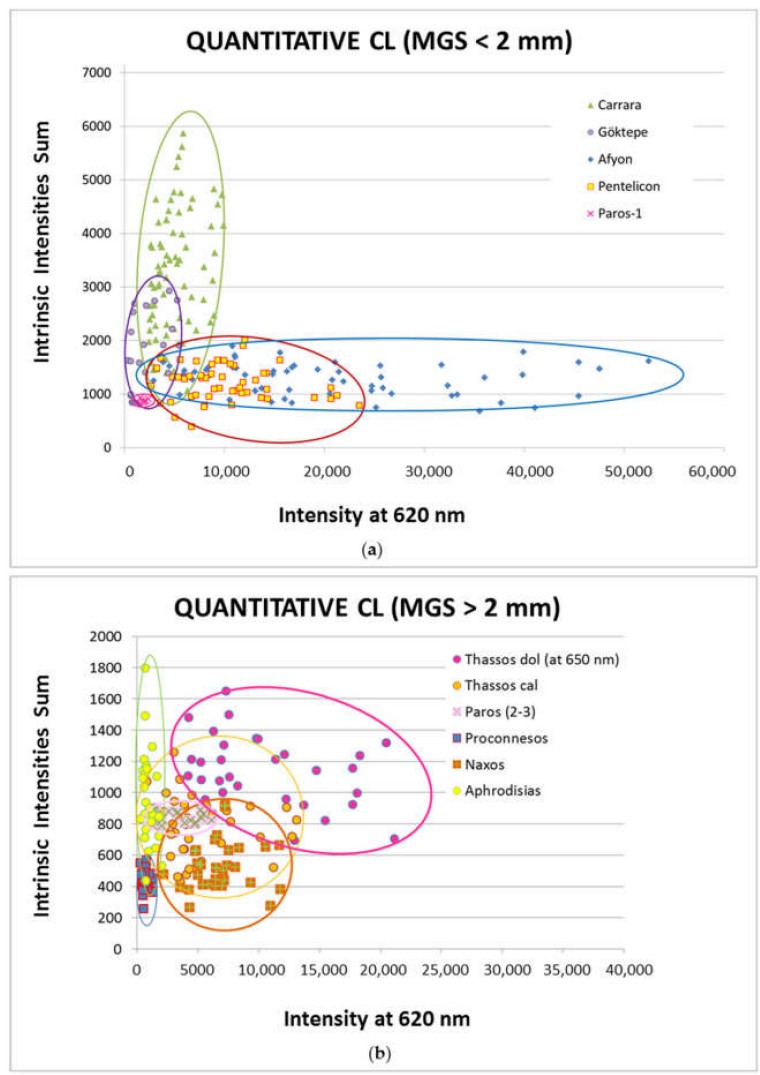

**Figure 8.** General quantitative CL diagrams for the main classical marbles according to their maximum grain size. The fields are 90% probability ellipses. (**a**) Diagram for fine-grained marbles; (**b**) diagram for medium to coarse-grained marbles.

3.2.4. Other CL Diagrams with Potential Use in Archaeometric Studies of Marble Provenance

To illustrate the potential usefulness of the data obtained in this study, we have confronted the quantitative CL values with other geochemical values measured in the same samples, taken from literature. As one example, Figure 10 shows the discrimination obtained using the C and O isotopic values for Göktepe [20] and those marbles from Carrara with also low extrinsic CL intensity (Fantiscritti) [30].

On the basis of mere observation of these diagrams, it is evident that this combination of data could contribute to identifying the provenance of a given marble. Other different confrontations of data are also possible to visualize the utility of the quantitative CL data since the marble samples studied here have been widely characterized by other methods.

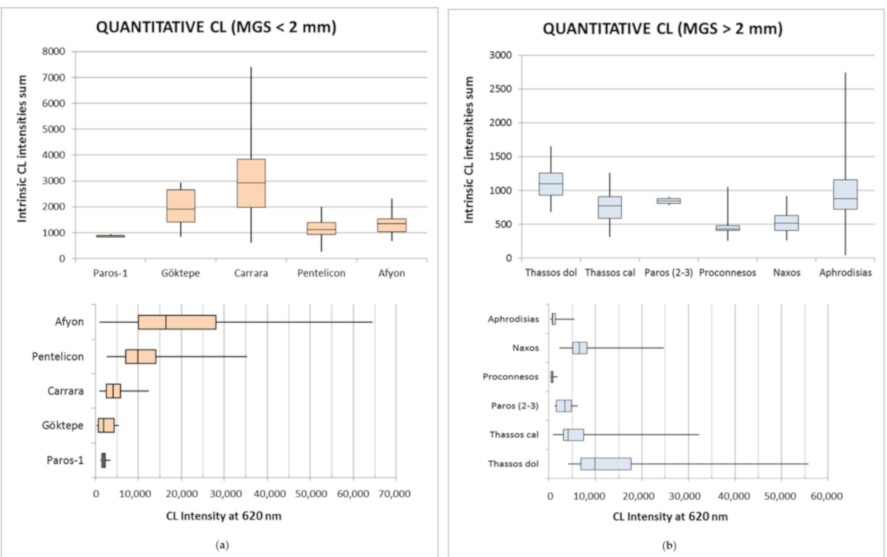

**Figure 9.** General quantitative CL boxplots and whiskers diagrams for the main classical marbles according to their maximum grain size. Medians, 25th and 75th percentiles, and min-max ranges are denoted by a central line, box, and whiskers, respectively. The extrinsic and the intrinsic CL are represented by the horizontal and the vertical boxes, respectively. In the whiskers lines are included outliers (**a**) Diagram for fine-grained marbles (MGS < 2 mm); (**b**) diagram for medium to coarse-grained marbles (MGS > 2 mm).

*3.3. Inter-Laboratory Comparison of CL-Images from the Main Quarry Marbles Used in Antiquity*

As it is well known that CL-imaging can produce slightly different results according to the different instrumentation used and the related setting [20,21], a selection of samples from the main classical marbles was the object of a complementary test to know which differences are obtained using two different cold CL-microscopes applied to the same thin-sections. The result is shown in Figure 11, where the identification sample of each thin-section is included.

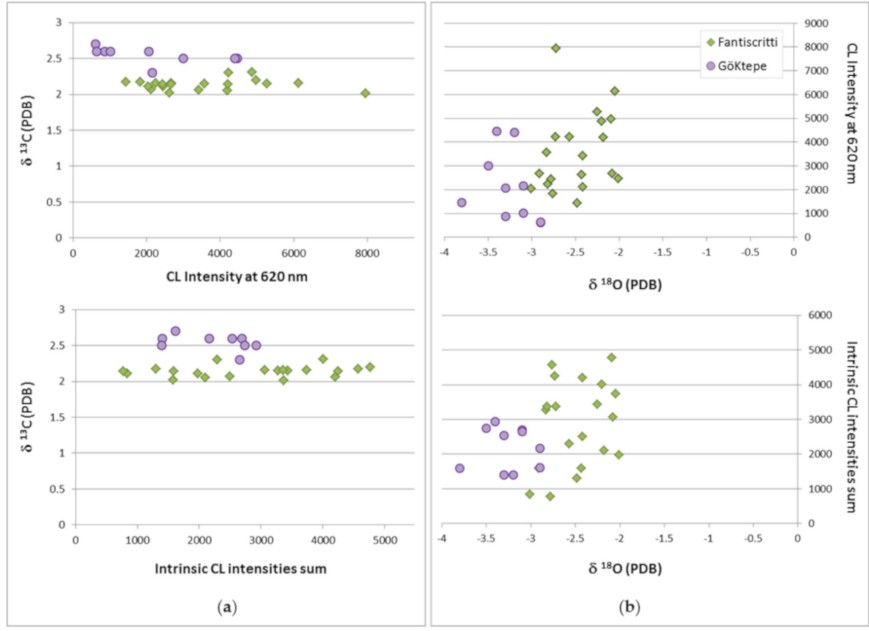

**Figure 10.** Different diagrams for Carrara (Fantiscritti) and Göktepe marbles using quantitative CL data obtained in this study and C and O stable isotopes from the literature [20,30]. (**a**) Two diagrams using quantitative CL versus $\delta^{13}$C; (**b**) two diagrams using quantitative CL versus $\delta^{18}$O.

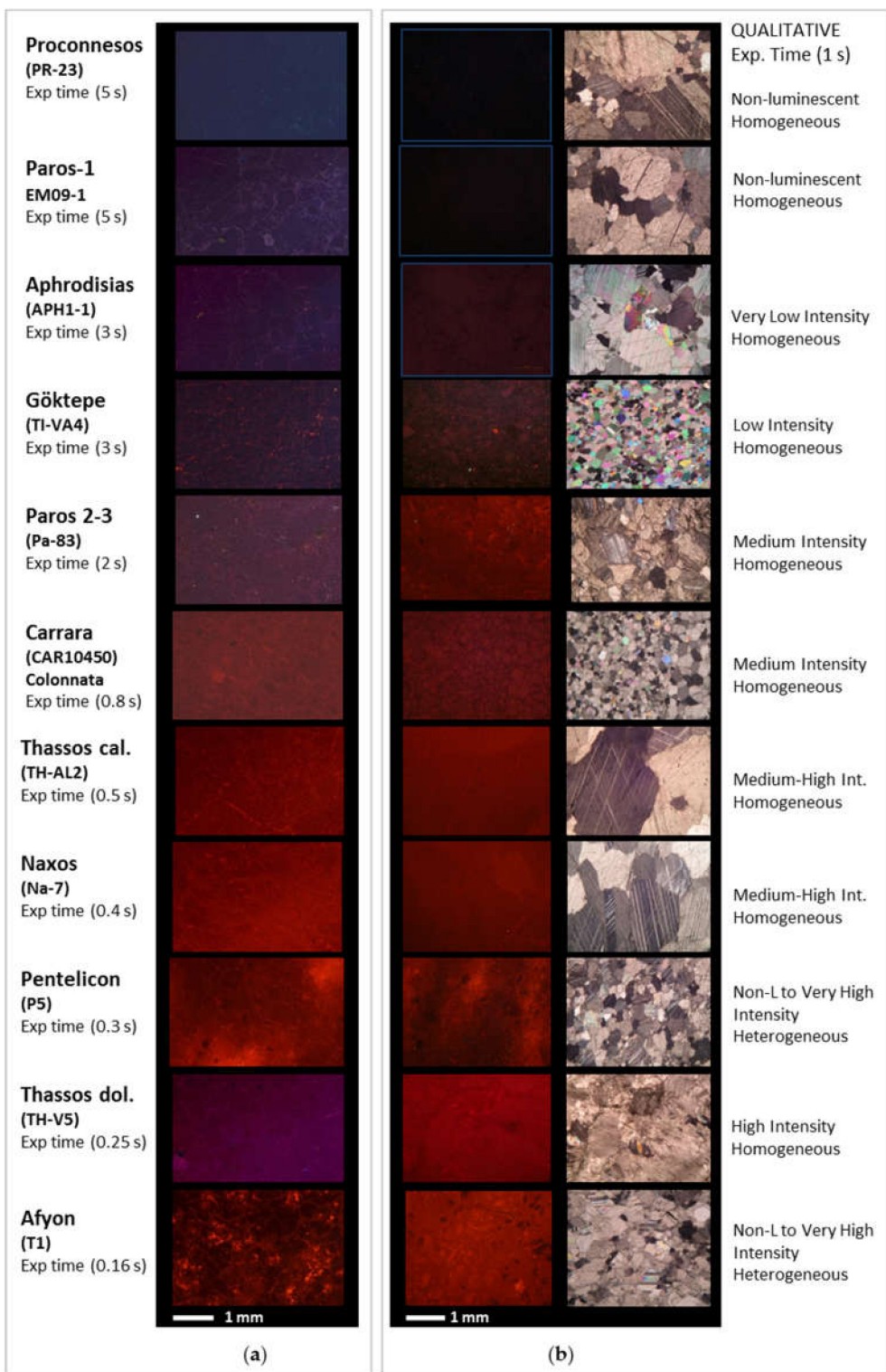

**Figure 11.** Inter-laboratory comparison of CL-images from the main white marbles used in ancient times. (**a**) CL-microphotographs taken at the Parisian laboratory of Pierre et Marie Curie, Sorbonne-University (vacuum 0.08 mbar, beam current 250–300 μA, electron energy 12–15 kV, 1/6 s to 5 s of exposure time, ISO 500). They are organized sequentially in decreasing order of the exposure time; (**b**) CL-images and the respective microphotograph in crossed polars of the same thin-sections taken at the laboratory of ICAC in Tarragona (vacuum 0.17 mbar, beam current 250–300 μA, electron energy 15–20 kV, automatically recorded at 1 s exposure time, 29 mm focal length, f/4.6 aperture, ISO-200).

Using the Cathodyn device of the laboratory Pierre et Marie Curie Sorbonne-University, and following the analytical conditions above stated (Chapter 2), several CL-images (ISO 500) were taken by different manual exposure time (from 5 s to 1/6 s) (Figure 11a). They are organized sequentially, from top to bottom, in order of decreasing exposure time.

Figure 11b shows the CL-images obtained by means of the optical CL device from the laboratory of the ICAC in Tarragona, applied to the same thin sections. The specific operational conditions have been above indicated (Chapter 2). The respective CL-patterns, automatically obtained in this case, with 1 s exposure time, are shown in the same order of Figure 11a, to facilitate their visual comparison. On the right side is included their respective qualitative CL description. In this database of CL-images from Tarragona, the sample shown for Carrara (CAR10450) is taken as the qualitative standard medium intensity.

Comparing the results from laboratory to another, it is evident that those taken in Paris have the advantage of better discrimination among those marbles with very low to low intensities, qualitatively characterized in the Tarragona images. Proconnesos and Paros−1 are especially well identified by their dark blue hue in Figure 11a, but they seem to be non-luminescent in Figure 11b.

Among the medium to coarse-grained marbles with very low intensity, Aphrodisias shows almost non-luminescent in the Tarragona CL-pattern, but slightly reddish after a meticulous polishing of the thin section with very-fine diamond grains which is shown in Figure 11b. However, by the Paris conditions, the same polished Aphrodisias sample exhibits quite similar dark blue pattern to that of Paros−1 marble (Figure 11a). Paros 2–3 shows brighter luminescent in both laboratories, although their respective hue is slightly different.

In white Göktepe marble, two petrographic varieties have been recently characterized with the aid of CL-patterns, each one with different CL intensity [18] and reconsidering the identifications previously made in archaeological pieces from Villa Adriana [29]. The sample shown in the CL-microphotographs of Figure 11a,b, is representative of the variety of low CL intensity, qualitatively assigned in literature [18]. Both CL-microphotographs, taken in their respective laboratory, display quite similar images.

Regarding other fine-grained marbles (MGS < 2 mm), that is Carrara, Pentelicon and Afyon, their CL-patterns are not very different in each laboratory. Although the CL-images shown for Afyon marble are quite different in Figure 11a,b, they are taken in the same thin-section, highlighting its typical CL heterogeneity very useful for its identification.

Thasos dolomitic marble is also better characterized in the images from Paris, since that of Tarragona exhibits a reddish tone close to that for calcitic marbles. Finally, Naxos and Thassos calcitic CL-microphotographs are very similar in both laboratories.

## 4. Conclusions

After the experimental test carried out on one marble sample to ensure the optimal operational process for obtaining the CL spectra and their correct measurements of intensities, this study provides the first quantitative CL database of the main quarry districts of the central and eastern Mediterranean. The 473 marble samples here analyzed belong to other published databases, 454 of them to the database published by Attanasio and collaborators [30] and the 19 Göktepe samples, to the databases recently published by Attanasio et al. and Wielgosz-Rondolino et al. [20,31]. As all samples have already been characterized by other analytical techniques, this increases the potential value of the quantitative CL data provided in this study. The full list of intrinsic and extrinsic CL data is available on the Tables S1–S10.

Marble samples of the main marble districts of the western Mediterranean are currently being subjected to the same analytical CL process in order to complete this quantitative CL database. Those districts already published [26], but also ancient quarries from the NW Iberia and the Pyrenees are taken into consideration.

Different diagrams expressing the differences and dispersion obtained for the UV and visible spectrum measurements have been discussed for each marble district. Taking into account the principal



petrographic parameter (MGS), two general quantitative CL diagrams have been proposed for future archaeometric studies of marble provenance. In addition, to illustrate the possible utility of combining the quantitative CL data with other usual parameters, other diagrams have been shown.

Applying optical CL microscopy to the study of ancient marbles, even being a qualitative technique sensitive to operational changes, is a helpful complementary technique for petrographic characterization, especially for Pentelicon and Afyon marbles. In this case, a well-organized quarry database with images taken under the same conditions enables its implementation. However, the comparison between the CL-microphotographs obtained in two different laboratories using the same thin-sections and two different equipment systems, highlighting the convenience of completing the usual qualitative CL observations with the quantitative measurements, like those reported here, and in particular for better discrimination among those with very low to low CL intensities and non-luminescent in the visible spectrum. Furthermore, the differences detected between the CL-images of both CL laboratories encourage us to test new analytical protocols in the Tarragona laboratory, especially for those very low luminescent samples to achieve the same degree of discrimination obtained in the laboratory of Paris.

Finally, it is worth noting that the multi-method investigation remains the most reliable approach for provenance discrimination and quantitative CL represents a further powerful contribution to assignments of unknown marble artifacts.

**Supplementary Materials:** The following are available online at http://www.mdpi.com/2075-163X/10/4/381/s1, Table S1: Quantitative CL database for 110 samples from the quarries of Carrara (Apuan Alps). Table S2: Quantitative CL database for 66 samples from the quarries of Afyon (Docimium, Iscehisar, Turkey). Table S3: Quantitative CL database for 19 samples from the quarries of Göktepe (Muğla, Turkey). Table S4: Quantitative CL database for 34 samples from the quarries of Aphrodisias (Turkey). Table S5: Quantitative CL database for 51 samples from the quarries of Proconnesos (Turkey). Table S6: Quantitative CL database for 58 samples from the quarries of Pentelicon (Greece). Table S7: Quantitative CL database for 29 samples from the quarries of Paros (Greece). Table S8: Quantitative CL database for 34 samples from the quarries of Naxos (Greece). Table S9: Quantitative CL database for 38 calcitic samples from the quarries of Thassos (Greece). Table S10: Quantitative CL database for 34 dolomitic samples from the quarries of Thassos (Greece).

**Author Contributions:** Conceptualization, software, and investigation, P.B. and M.P.L.M.; methodology, P.B. and M.P.L.M.; analysis and data curation, P.B., M.P.L.M., and A.G.G.-M.; writing—Original draft preparation, P.B. and M.P.L.M.; writing—Review and editing, M.P.L.M.; funding acquisition, A.G.G.-M. and M.P.L.M. All authors have read and agreed to the published version of the manuscript.

**Funding:** This research was partially funded by the Spanish Government, Ministerio de Ciencia, Innovación y Universidades, and is part of the objectives of the project 'Officinae lapidariae Tarraconenses. Canteras, talleres y producciones artísticas en piedra de la Provincia Tarraconensis (HAR2015-65319-P, MINECO/FEDER, UE)', with the additional collaboration of the project 'El mensaje del mármol: prestigio, simbolismo y materiales locales en las provincias occidentales del imperio romano (PGC2018-099851-A-I00, MINECO/FEDER, UE)'.

**Acknowledgments:** The authors would like to express their gratitude to D. Attanasio, M. Brilli, and M. Bojanowski for providing samples to carry out this study; to the reviewers for their constructive and useful remarks; to O. Boudouma, from the Institut des Sciences de la Terre de Paris (iSTeP) Sorbonne Université, for measuring the CL spectra and to H. Royo, specialist technician at the ICAC (2011–2017), for his assistance on the CL-imaging; and to acknowledge the use of Servicio General de Apoyo a la Investigación-SAI, Universidad de Zaragoza, for the technical support.

**Conflicts of Interest:** The authors declare no conflict of interest.

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
