# Peer review of "A New Database of the Quantitative Cathodoluminescence of the Main Quarry Marbles Used in Antiquity"

_minerals, doi:10.3390/min10040381_

Round 1
Reviewer 1 Report
The manuscript entitled ‘A Complete New Database of the Quantitative Cathodoluminescence of Quarry Marbles Used in Antiquity’ presents an interesting methodological study for the provenance of white marbles. In the manuscript, the authors wish to (1) present a complete quantitative CL database of the main marbles from Classical Antiquity, (2) explore the effect of Mn on CL in marble, and (3) compare CL-microfacies between laboratories.
The methodology and techniques are scientifically correct, innovative and present a very valuable contribution to marble characterisation and the field of marble provenancing. The authors are clearly well-acquainted with the subject, methods and techniques. Both methodology and results are accurately described.
I suggest some clarifications and additions in the data treatment and data presentation.
(1) The authors use the slope of the line between Q1, Q2 and Q3 to show which of the spectral intensities (intrinsic or extrinsic) presents the largest dispersion. Slope is also used to fingerprint or differentiate between quarries. I suggest the authors to explain why slope based on the quartiles was chosen as parameter, and why not other statistical methods were used that are based on all sample observations (e.g., linear regression), or why the data ellipses or data fields were chosen for representation (similar to the standard method for stable isotopic analysis). In line with this remark, the authors should consider adding the analytical results for each sample as an appendix to the article. In combination with other recent studies on marble characterisation where analytical data is presented, this would greatly increase reproducibility, verifiability and value of the research for the scientific community, and will promote the reuse and extension of the methodology (in line with the open data, open science movement).
(2) The authors state in the conclusions that ‘Quantitative CL and CL-microfacies provide complementary parameters for the characterization of the “finger prints” of marbles used in antiquity, which should be always used as a complementary to the petrographic features. They are particularly valuable when other techniques are not diagnostic, especially when isotopic overlapping occurs.’ It would be nice to see this applied to a particular archaeological case study to illustrate the potential and drawbacks of the research. Judging from the summarised CL-results, it seems that, like in stable isotopic analysis, there is considerable overlap between the different quarry districts and that there can be problems with outlier-values. Similar to remark 1, it would be useful to see the effect of using the summarised results versus all sample observations.
In conclusion, the manuscript can be considered a very interesting methodological study about ancient marble characterisation. The research is scientifically and experimentally correct for the techniques used. Overall, after few minor changes and additions (see above and below), I recommend publication of the manuscript in the journal “Minerals”.
Below is listed the points related to language that benefit from modification.
| L146: | Replace 'de' by 'the' |
| L391 | Replace 'finger print' by 'fingerprint' |
Author Response
Please, see the attachment "minerals-770320_Clarifications and additions suggested by Reviewers.pdf

Reviewer 2 Report
This is an interesting and potentially valuable work that concerns the application of quantitative cathodoluminescence (CL) analysis to marbles exploited in the most important ancient quarries in the central and eastern Mediterranean. The approach applied is based on measurements of the intensity of UV (intrinsic emission) and visible light (extrinsic emission) spectra emitted by marbles when bombarded with electrons released by a cathode. The equipment employed is a scanning electron microscope (SEM) coupled to a CL spectrometer. The intrinsic CL is related to defects in the crystal structure, whereas the extrinsic CL to the relative abundance of activators (mainly Mn2+) to quenchers (mainly Fe2+) in the carbonate crystal lattice. The main results of this research are generalized statistic parameters related to the distribution of both emission spectra for the ten most important ancient marble quarry areas. These data represent the first database of quantitative CL properties of these marbles. They are presented in a table and as binary diagrams plotting the extrinsic vs. intrinsic CL intensity of spectrum recorded at 620 nm characteristic for calcite. The statistical parameters presented are the first quartile (Q1), the third quartile (Q3) and the median (Q2). According to the authors, these data "will be helpful for future applications in the identification of marble used for archaeological pieces", which appears to represent the main goal of this work. Additionally, this paper analyzes the relation between the extrinsic CL intensity and Mn content in calcite and the difference in CL pattern observed with two different optical CL instruments.
Major issues
- These research questions are sound and the methods used to answer them appear suitable. The research material is properly chosen, as these quarries were the major sources of marble during Hellenistic and Roman periods and they exhibit a wide range of CL patterns. The number of samples analyzed is also statistically valid, but there is a large difference between the most (Carrara - 110 samples) and least represented quarries (Göktepe - 20 samples). Such significantly larger number of samples analyzed for Carrara may have affected the statistics to some extent, e.g. by increasing the dispersion of values measured for this marble. The statistical parameters would be more comparable if the number of measurements was more similar for all quarries. With regard to the number of samples analyzed, Carrara is in the first place with 110 samples, which is 2/3 more than for the second Afyon quarries with 66 samples. Moreover, there is a great variability in the CL intensities among the Carrara samples related to the location (see Fig. 3). Therefore, I suggest to split Carrara samples into three sub-groups, which may solve this and other problems as well (see below).
- Methodology applied appears correct and properly described. One of the main flaws of this work is that the true data is not presented for all quarries. What is provided are only statistical parameters, which, as I explain below, are not the best way to present the data. I can imagine, that extrinsic and intrinsic CL spectra intensities can be presented for all 477 samples as a supplementary material, especially in an open-source journal. However, I think that the most valuable way of presenting the data would be the standard box & whiskers plots, which would show the following values: minimum, maximum, 1st quartile, 3rd quartile, median, outliers (if any). There would be two such diagrams: one for intrinsic and one for extrinsic CL intensity. Presenting only Q1 to Q3 excludes half of the actual data obtained for a given quarry (those between minimum value and Q1 plus those between Q3 and maximum value). Such manipulations should be avoided, because they oversimplify the real data and are not really representative of the dataset.
- The problem with the statistical parameters used in the current version of the manuscript is that they represent a considerable generalization, which obscures the real picture and makes the potential applicability of this database much lower than if box & whiskers plots were used. I cannot see how is the current presentation of the database supposed to make provenancing studies of marble artifacts easier? Let's say that I am to identify the source of an artifact and I perform quantitative CL, apart from other methods, of course. I will obtain actual intensity values (for extrinsic and intrinsic CL). How am I supposed to say which quarry the values obtained for this artifact correspond to if I have only the Q1, Q2 and Q3 values generalized for each of the ten quarries given in the database? This must be explained by the authors or the suggested box & whiskers plots are provided. With such presentation, I would be able to compare the values obtained for the artifact with those reported for the classical marble quarries in the database.
- On Fig. 3 it can be observed that there is a strong dispersion of projection points for Carrara, which has been noted by the authors, of course. The three points representing the quartiles in this diagram are perfect illustration of how huge generalization results from such statistical approach. After having a closer look at the diagram, I observed that some quarries exhibit similar distribution of projection points, which is very likely related to similar CL patters. Therefore, I suggest to make an attemps to split the data into three sub-groups: (1) Polvaccio + Ravaccione + Canalgrande; (2) Sponda + Fantiscritti; (3) Fossacava + Calagio + Gioia, and to analyze the data treating them as separate quarries, as it is often the case with regard to other areas, such as Paros or Thasos that exhibit variable CL patterns depending on actual quarry (Paros-1, Paros-2 etc.). I would very much like to see box & whiskers plots separately for these Carrara, Paros and Thasos sub-groups.
- This large dispersion of projection points for Carrara indicates also that CL patterns may be very different between samples from Carrara. For instance, I have studied samples from Fantiscritti, which were non-lumienscent or with extremely low-intensity orange CL, which is incomparably weaker CL than that of Carrara samples characterized by 620 nm intensity > 10000, which are reported in the reviewed manuscript. Therefore, treating Carrara marble as a standard for optically-judged moderate CL intensity, as stated in lines (185-187), without indicating from which particular quarry it was derived, is not reliable.
- I have not studied CL of Docimian marble yet. However, based on extensive fieldwork, detailed sampling and a large number of stable C and O isotope measurements, I can say that properties of Docimian marble change dramatically between its quarries, which are dispersed in a very large area. Therefore, it is not surprising that even larger dispersion of projection points is observed for Docimium (Fig. 4a) than for Carrara. Similar to Carrara, it would be valuable to make an attempt and split the samples into several subgroups depending on the quarry location. I imagine, that the authors may not possess precise locational data for the samples. If so, this approach cannot be applied in the current work.
- There is one more advantage of the quantitative CL applied in this work, which could be emphasized. If I understand correctly, marbles with dark blue luminescence color are characterized by high intrinsic and very low extrinsic CL (e.g. Proconnesos, Paros-1), whereas those that are non-luminescent exhibit very low both kinds of CL. Marbles with high extrinsic CL show vivid orange CL colors and in such cases the intrinsic CL cannot be judged by the optical qualitative CL analysis. The advantage of quantitative SEM-CL is that the intrinsic CL can be measured even is the intensity of extrinsic CL is high.
- How can the authors explain the difference in extrinsic CL intensity between Göktepe samples provided by the two research teams? I have collected with my own hands all samples used in Wielgosz-Rondolino et al. (2020) and there were often problems with obtaining fresh, not weathered samples. Such deteriorated samples were discarded. What about the samples used in Brilli et al. (2018)? I wonder if the preservation state of marbles may affect their extrinsic CL intensity?
- Based on what I can see in Fig. 7, I disagree with the interpretation provided in lines 270-272 that the values of Göktepe are much lower than those of Carrara. Since the uppermost projection point of Göktepe is its Q3, whereas the lowest point of Carrara is its Q1 and the projection point for median values of Göktepe is above Q1 of Carrara, it must be concluded that there is a large overlap between Göktepe and Carrara encompassing slightly more than 50% of Göktepe values.
- The relationship between extrinsic CL intensity and Mn concentration discussed in lines 317-323 and depicted in Fig. 8b are based on confrontation of data, which were obtained for different samples. What the authors did is a valuable attempt, which, however, was based on incomparable datasets. This should be tested on the same samples. I suggest that the authors select thin sections corresponding to the samples analyzed with SEM-CL and perform in-situ elemental analyses with the use of laser ablation or electron microprobe. Only such approach can provide direct correlation between Mn concentration and extrinsic CL intensity. Moreover, the authors included median values calculated for five quarries (Paros, Göktepe, Carrara, Pentelikon, Afyon) analyzed with LA-ICP-MS by Poretti et al. (2017). Such values provide a generalization, whereas marbles from these quarries, Pentelikon and Afyon in particular, exhibit strong variability of extrinsic CL intensity, as shown in Fig. 7 in the reviewed manuscript.
- It could be stressed that quantitative SEM-CL is a potential solution to problems related to interlaboratory bias described in lines 349-362.
Minor issues
A considerable part of minor issues that I encountered is related to language problems, so they are not scientific flaws. Language correction is advised. Other minor problems are commented below.
- 44-47: Quenchers, such as Fe2+ should also be mentioned as factors influencing CL pattern.
- 49-51: In my opinion, the meaning of "CL microfacies" is closer to "CL pattern" than to "CL microphotographs" or "CL images", as it is suggested in this sentence and used throughout the manuscript. The term "microfacies" is derived from carbonate sedimentology, where it refers to textural and compositional characteristics of a sedimentary carbonate rock. I strongly suggest to use "CL microphotographs" or similar expression. Otherwise, it is confusing.
- 67: This database is surely not complete, as many other quarries wait to be investigated with this method.
- 71-72. I believe that a similar interlaboratory comparison was made by Wielgosz-Rondolino et al. (2020).
- 86: What does 1 second refer to?
- 143: ...is less significant...
- 143-144: I believe that this of 620 nm position is measurable as well, although the difference is less significant.
- 144-145: This sentence is a repetition. It has already been written above.
- 151: Does the strength of vacuum influence the results? If so, vacuum strength must be given exactly, so that other laboratories can use the same parameters and produce comparable results.
- 154-156: unclear
- 159-161: The median value (Q2) does not necessarily represent the central value. The central value is represented by average.
Fig. 3: What are the short horizontal lines below the black solid circles indicating Q1, Q2 and Q3?
- 326-328: unclear
Fig. 8a: part of the caption unclear: "with indication of those evaluated in [35]"
- 360-361 and Fig. 10 for Paros: What is similar in both laboratories? Which is which in this sentence and Fig. 10a? Paros-1 is on the left and Paros-2-3 on the right or the opposite? Please, make it clear. And which Paros is in Fig. 10b?
- 370: Is there black cathodoluminescence? I always thought that such samples are referred to as "non-luminescent"?
- 390: Instead of "CL-microfacies" use "cathodoluminoscopy" or "CL-imaging" in that particular case.
- 408: Thank you for acknowledging my modest help in your research. Please note that my name is spelled Bojanowski.
Figures are generally clear and informative, although the plots could be improved by making the numbers along axes larger. The structure of the manuscript is clear and logical.
In summary, this work is interesting with good quality data, but their statistical elaboration is inadequate, which hampers the potential applicability of this contribution. Additional analyses are needed for reliable analysis of the relationship between Mn concentration and the extrinsic CL intensity.
I hope that my comments are helpful and that it is possible to make this contribution even more valuable,
Author Response
Please, see the attachment "minerals 770320_Clarifications and additions suggested by Reviewers.pdf

Round 2
Reviewer 1 Report
In comparison to the first version of the manuscript, the suggested changes have been applied by the authors based on the reviewers’ suggestions. In relation to the content, the manuscript needs some corrections to the data in table 1. The numbers and statistics represented in table 1 are not correct when working with the data from the supplement. Similarly, the total number of samples analysed are not correct (472 listed in the supplement, and not 476 as indicated in the text and table 1). Below are the correct values based on the supplement. I suggest correcting either the supplement or table 1 (and the values in the text), depending on whether the errors are in the supplement or the manuscript. Please make sure no rounding is applied in the text and table and that the values in the table and the text correspond (see, for example, Line 222 > 7416 and Table 1 > 7406 (for Carrara), or Line 244 > y = 0.041x and Table 1 > y = 0.0418x (for Afyon)).
Line 183-188: It needs to be clarified how the slope value was calculated. I was unable to reproduce the results presented in the manuscript based on the database.
For some newly added sections, language and text editing is advised.
| L345, 388 416, 530 | 'fine grained' becomes 'fine-grained' |
| L202, 389, 398, 417, 518 | 'medium and coarse grained' becomes 'medium- and coarse-grained' |
| L521 | 'similar than' becomes 'similar to' |
| L522 |
The other Parian marble from Chrorodaki area (Paros 2-3), shows brighter luminescent in both laboratories, although their respective hue are is slightly different. |
|
L525-529 |
unclear |
|
L580-582 |
The 476 marble samples here analyzed belong to other published databases, 457 of them to that the database published by Attanasio and collaborators [30] and the 19 Göktepe samples, to those the databases recently published by Attanasio et al. and Wielgosz-Rondolino et al. [20,31]. |
|
L582 |
delete 'Therefore' |
|
L585 |
Marble samples for of the main marble districts of the Western Mediterranean are currently being subjected to the same analytical CL process in order to complete this quantitative CL database. These include including, not only those districts already published [26] and now updated to the same operational conditions, but also ancient quarries from the NW Iberia and the Pyrenees, are being subjected to the same analytical CL process in order to complete this quantitative CL database. |
|
L599 |
Add comma after 'However' |
Table 1 corrections:
|
quarry |
n |
min |
Q1 |
Q2 |
Q3 |
max |
min |
Q1 |
Q2 |
Q3 |
max |
a-value1 |
a-value2 |
|
afyon |
66 |
864 |
10591.5 |
16374.5 |
26519.75 |
64511 |
682 |
1057.5 |
1360 |
1530 |
2321 |
0.029664 |
0.0418 |
|
thasos-dol |
34 |
4201 |
6945 |
9858.5 |
17141.5 |
55733 |
684 |
937.5 |
1104.5 |
1241.75 |
1651 |
0.029839 |
0.028 |
|
pentelikon |
58 |
2668 |
7126.5 |
9864.5 |
13870.25 |
35231 |
269 |
945.25 |
1118.5 |
1373.5 |
2007 |
0.063503 |
0.686 |
|
naxos |
34 |
2209 |
5219 |
6562.5 |
7947.75 |
24665 |
265 |
410.25 |
517 |
628.25 |
921 |
0.07989 |
0.539 |
|
carrara |
110 |
939 |
2540.25 |
4071 |
5778 |
12475 |
618 |
1978.25 |
2924 |
3799.5 |
7407 |
0.562505 |
0.5484 |
|
thasos-cal |
38 |
788 |
3158.75 |
4065 |
7234.75 |
32298 |
316 |
604 |
771.5 |
904.5 |
1259 |
0.073724 |
0.046 |
|
paros2(3) |
20 |
1183 |
1606.5 |
3348.5 |
4758.25 |
6155 |
786 |
809.75 |
843.5 |
875.25 |
903 |
0.020782 |
0.2067 |
|
goktepe |
19 |
353 |
684 |
1919 |
4150.5 |
5504 |
848 |
1496 |
1919 |
2593 |
2928 |
0.316458 |
0.582 |
|
paros1 |
8 |
1195 |
1597.25 |
1821.5 |
2006.75 |
2369 |
832 |
848.5 |
871.5 |
892.75 |
937 |
0.108059 |
0.4678 |
|
aphrodisias |
34 |
320 |
632.25 |
744.5 |
1281 |
5510 |
43 |
723.25 |
878 |
1150.75 |
2748 |
0.65896 |
0.3791 |
|
proconnesos |
21 |
277 |
490 |
618 |
798.5 |
1867 |
259 |
410.5 |
433 |
484 |
1058 |
0.23825 |
0.5207 |
a-value1 = based on ‘Q3/Q1’, a-value2 = based on ‘max/min’.
Author Response
1. The numbers and statistics represented in table 1 are not correct when working with the data from the supplement.
Data in the supplement was revised and one forgotten sample (NY5) was added now at Paros-1. This sample was included in our calculations, but not in the slope. The rest of numbers was OK in the Supplementary Tables. Definitely, in total are 473 samples.
The statistics were obtained based on those 473 samples (included outliers). The differences obtained by reviewer must be due to the program used. It seems to be that the reviewer uses a program which find first the outliers and they are not taken into account in the statistics, especially in Q1 and Q3 values. In fact, Q2 is practically equal in the numbers obtained by reviewer and those obtained by authors (as can be seen in the comparison of values in the table below). In any case, our data have been revised, we have used Excel 2010 version, in which outliers are not automatically calculated. To check the minimal differences found, I write our numbers in the same Table provided by reviewer 1, each value different below your value;
Another minimal difference is related to the decimals: which were first not taken into account, but now we have prorated adding a unit when the decimal is> 5. In bold are those numbers now different from the previous submission.
2. Similarly, the total number of samples analysed are not correct (472 listed in the supplement, and not 476 as indicated in the text and table 1).
There was a mistake with one sample, as mentions above, so definitely are 473 samples. Yes, the mistake has been emended.
3. Below (Table) are the correct values based on the supplement.
Sorry, but they are your “correct values”, but I do not know how do you calculated them. Probably, your data are also correct if the outliers are not in consideration. If all data (with the exception of the one missing in Paros-1) are the same, the differences must be sought in the way to obtain them. In our case we understand that it is better to take into account all values, including outliers, in fact if you compare the values in the Table below, your values are more restrictive in the Interquartile Range (Q1 to Q3), than our values.
4. I suggest correcting either the supplement or table 1 (and the values in the text), depending on whether the errors are in the supplement or the manuscript.
Corrections in Table 1 are made (number of samples, which were three mistakes, Carrara, Afyon and Proconnesos, those in bold in table below.
Concerning data, decimals have been rounded, as mentioned above (are also in yellow, in the .pdf).
5. Please make sure no rounding is applied in the text and table and that the values in the table and the text correspond (see, for example, Line 222 > 7416 and Table 1 > 7406 (for Carrara) or Line 244 > y = 0.0418x and Table 1 > y = 0.0418x (for Afyon))
OK, now 7407, with rounded decimal
6. Line 183-188: It needs to be clarified how the slope value was calculated. I was unable to reproduce the results presented in the manuscript based on the database.
Probably you write this sentence before you write the data in your Table.
Anyway, a-Values are calculated automatically by the program (Excel 2010) based on all data in each quarry representation. This is written on L186-187.
7. For some newly added sections, language and text editing is advised.
All have been emended
|
L345, 388 416, 530 |
'fine grained' becomes 'fine-grained' OK |
|
L202, 389, 398, 417, 518 |
'medium and coarse grained' becomes 'medium- and coarse-grained' OK |
|
L521 |
'similar than' becomes 'similar to' Ok |
|
L522 |
The other Parian marble from Chrorodaki area (Paros 2-3),shows brighter luminescent in both laboratories, although their respective hue are is slightly different. |
|
L525-529 |
Unclear. Changed |
|
L580-582 |
The 476 marble samples here analyzed belong to other published databases, 457 of them to that the databasepublished by Attanasio and collaborators [30] and the 19Göktepe samples, to those the databases recently published by Attanasio et al. and Wielgosz-Rondolino et al. [20,31]. |
|
L582 |
delete 'Therefore' |
|
L585 |
Marble samples for of the main marble districts of the Western Mediterranean are currently being subjected to the same analytical CL process in order to complete this quantitative CL database. These include including, not only those districts already published [26] and now updated to the same operational conditions, but also ancient quarries from the NW Iberia and the Pyrenees, are being subjected to the same analytical CL process in order to complete this quantitative CL database. |
|
L599 |
Add comma after 'However' |
8. Table 1 corrections
You are right with the number of samples in Afyon and in Carrara, but not in Proconnesos. All changes made now are in yellow, I write our data to compare to yours.
|
quarry |
n |
min |
Q1 |
Q2 |
Q3 |
max |
min |
Q1 |
Q2 |
Q3 |
max |
a-value2 |
|
|
afyon |
66 |
864 |
10591.5 10038 |
16374.5 |
26519.75 27990 |
64511 |
682 |
1057.5 1044 |
1360 |
1530 1534 |
2321 |
0.0418 |
|
|
thasos-dol |
34 |
4201 |
6945 6877 |
9858.5 |
17141.5 17701 |
55733 |
684 |
937.5 930 |
1104.5 |
1241.75 1258 |
1651 |
0.028 |
|
|
pentelikon |
58 |
2668 |
7126.5 7000 |
9864.5 |
13870.25 13992 |
35231 |
269 |
945.25 941 |
1118.5 |
1373.5 1383 |
2007 |
0. 0686 |
|
|
naxos |
34 |
2209 |
5219 5153 |
6562.5 |
7947.75 8164 |
24665 |
265 |
410.25 408 |
517 |
628.25 630 |
921 |
0. 0539 |
|
|
carrara |
110 |
939 |
2540.25 2523 |
4071 |
5778 5789 |
12475 |
618 |
1978.25 1971 |
2924 |
3799.5 3849 |
7407 |
0.5484 |
|
|
thasos-cal |
38 |
788 |
3158.75 3086 |
4065 |
7234.75 7427 |
32298 |
316 |
604 584 |
771.5 |
904.5 906 |
1259 |
0.046 |
|
|
paros2(3) |
20 |
1183 |
1606.5 1603 |
3348.5 |
4758.25 4799 |
6155 |
786 |
809.75 809 |
843.5 |
875.25 878 |
903 |
0.2067 |
|
|
goktepe |
19 |
353 |
684 639 |
1919 |
4150.5 4399 |
5504 |
848 |
1496 1406 |
1919 |
2593 2655 |
2928 |
0.582 |
|
|
paros1 |
9 |
1195 |
1597.25 1571 |
1821.5 1918 |
2006.75 2270 |
2369 3513 |
832 |
848.5 839 |
871.5 |
892.75 899 |
937 |
0.3965 |
|
|
aphrodisias |
34 |
320 |
632.25 616 |
744.5 |
1281 1343 |
5510 |
43 |
723.25 720 |
878 |
1150.75 1158 |
2748 |
0.3791 |
|
|
proconnesos |
1 51 |
277 |
490 484 |
618 |
798.5 820 |
1867 |
259 |
410.5 |
433 |
484 |
1058 |
0.5207 |
a-value1 = based on ‘Q3/Q1’, a-value2 = based on ‘max/min’.
a-value = based on all samples (range from min to max)

Reviewer 2 Report
The work has been significantly improved. I have made some minor corrections and provided some comments in the annotated pdf file (attached). These are mostly related to language problems.

Author Response
Please, see the attachment
